# Why Do Adolescents Skip Breakfast? A Study on the Mediterranean Diet and Risk Factors

**DOI:** 10.3390/nu17121948

**Published:** 2025-06-06

**Authors:** Cristina Romero-Blanco, Evelyn Martín-Moraleda, Iván Pinilla-Quintana, Alberto Dorado-Suárez, Alejandro Jiménez-Marín, Esther Cabanillas-Cruz, Virginia García-Coll, María Teresa Martínez-Romero, Susana Aznar

**Affiliations:** 1PAFS Research Group, Department of Nursing, Physiotherapy and Occupational Therapy, Ciudad Real Faculty of Nursing, University of Castilla-La Mancha, 13071 Ciudad Real, Spain; cristina.romero@uclm.es; 2PAFS Research Group, Faculty of Sports Sciences, University of Castilla-La Mancha, 45071 Toledo, Spain; ivan.pinilla@uclm.es (I.P.-Q.); alberto.dorado@uclm.es (A.D.-S.); alejandro.jimenez16@uclm.es (A.J.-M.); esther.cabanillas@uclm.es (E.C.-C.); virginia.garcia@uclm.es (V.G.-C.); susana.aznar@uclm.es (S.A.); 3Department of Physical Activity and Sport, Faculty of Sport Sciences, Campus of Excellence Mare Nostrum, University of Murcia, 30720 Murcia, Spain; mariateresa.martinez13@um.es

**Keywords:** skipping, breakfast, adolescent, diet, Mediterranean, overweight, adolescent, obesity, adolescent

## Abstract

**Background/Objectives**: Skipping breakfast is increasingly common among adolescents and has been associated with adverse health and academic outcomes. The average prevalence of breakfast skipping among adolescents is around 16%, although worldwide, it varies greatly across studies, ranging from 1.3 to 74.7%. This study aimed to assess the frequency of daily breakfast consumption and explore the factors associated with its omission in a stratified sample of Spanish adolescents. **Methods**: A cross-sectional study was conducted among 547 third-year secondary school students (aged 14–15) from both urban and rural areas in Castilla-La Mancha. Self-reported questionnaires were used to gather sociodemographic, psychosocial, and lifestyle data, including adherence to the Mediterranean diet (via the Kidmed questionnaire) and breakfast habits during school days. Descriptive, bivariate (Chi-square), and multivariate (binary logistic regression) analyses were conducted separately for boys and girls. **Results**: Findings showed a high prevalence of breakfast skipping one or more days (33.46%), with a significantly higher rate among girls (43.27%) than among boys (24.42%). Also, girls were more likely than boys to skip breakfast every day (14.18% vs. 6.87%, *p* < 0.001). In both groups, low adherence to the Mediterranean diet was strongly associated with skipping breakfast, along with higher screen time, shorter sleep duration, and being overweight/obese. Among girls, low olive oil consumption (OR 0.145 (CI 0.03–0.67) *p* 0.014) and poor Mediterranean diet adherence (OR 0.140 (CI 0.06–0.34) *p* < 0.001) were significant predictors. For boys, being overweight/obese (OR 2.185 (CI 1.06–4.52) *p* 0.035), low Mediterranean diet adherence (OR 0.136 (CI 0.06–0.32) *p* < 0.001), and not eating industrial pastries were associated factors (OR 0.413 (CI 0.20–0.88) *p* 0.022). Predictive models demonstrated good discriminatory power (AUC = 0.807 for girls; 0.792 for boys). **Conclusions**: Skipping breakfast is prevalent among adolescents, particularly girls, and is linked to poor dietary patterns and excess weight. These findings underscore the need for gender-specific nutritional interventions to promote regular breakfast consumption and improve dietary habits in adolescents.

## 1. Introduction

Breakfast has traditionally been considered the most important meal of the day, providing the energy and nutrients necessary for optimal physical and cognitive performance [1]. However, the prevalence of breakfast skipping is increasing, especially among adolescents [2,3]. This trend is concerning, as several studies have associated breakfast omission with a series of adverse health outcomes, including increased risk of overweight and obesity, unfavorable lipid profiles, high blood pressure, and poorer academic performance [2,3,4,5,6,7].

Several hypotheses have been proposed to explain the relationship between skipping breakfast and adverse health outcomes. One theory, supported by various studies, suggests that skipping breakfast may lead to increased energy intake throughout the remainder of the day, contributing to weight gain [8]. Another hypothesis posits that those who skip breakfast generally have a lower-quality diet, with reduced intake of fruits, vegetables, and other nutrient-dense foods [9]. Accordingly, greater adherence to the Mediterranean diet is associated with regular breakfast consumption. There is evidence indicating that adolescents who eat breakfast at least six times per week higher adherence to the Mediterranean diet than those who skip it or eat it irregularly [10].

The percentage of adolescents who skip breakfast varies widely across studies, ranging from 1.3% to 74.7%, although most estimates fall between 10% and 30%, with an average of 16% [3]. This discrepancy is largely due to the different definitions used to conceptualize breakfast omission [2,7,11]. In Spain, the latest available data from the Health Behaviour in School-aged Children (HBSC) study report that approximately 20% of Spanish adolescents skip breakfast on weekdays, with higher rates among girls than boys [12]. This national trend reinforces the relevance of analyzing the determinants of breakfast skipping in Spanish youth.

With respect to gender, girls are generally more likely to skip breakfast [2,13], and this pattern often emerges even before adolescence [14]. Several explanations have been proposed for this disparity. Psychosocial factors, such as body image concerns and dieting behaviors, are more prevalent among adolescent girls and may contribute to intentional breakfast skipping [8]. In addition, hormonal fluctuations and appetite regulation differences may play a role [15]. From a behavioral perspective, girls may face greater morning time constraints related to grooming or school preparation routines, leading them to deprioritize breakfast [13]. Nutritionally, girls who eat breakfast regularly tend to choose healthier options, such as dairy products or fruits, whereas those who skip breakfast are more likely to exhibit overall poorer dietary patterns [9]. Cultural or regional influences may also shape gender-specific breakfast behaviors [10]. These gender-based differences underscore the importance of investigating breakfast omission through a sex-disaggregated lens to identify specific intervention targets.

Although most research suggests a positive association between skipping breakfast and increased obesity risk, the causal relationship remains unclear and warrants further investigation [3,8]. It has been observed that children and adolescents who skip breakfast are more likely to be overweight or obese compared to their counterparts who eat breakfast regularly. However, some studies, such as that by López-Gil et al. [2], did not find a significant association between breakfast omission and overweight or identified gender-specific associations—for instance, observing the effect only among boys. Various mechanisms have been proposed to explain this potential relationship.

Skipping breakfast may disrupt metabolism and appetite-regulating hormones, leading to alterations in hunger and energy intake later in the day. One key mechanism involves the hormone ghrelin, which stimulates appetite and increases when fasting is prolonged, as is the case when breakfast is omitted. At the same time, levels of leptin—responsible for promoting satiety—may decrease, impairing satiety signaling and encouraging overeating during subsequent meals [15]. Additionally, skipping breakfast has been associated with decreased insulin sensitivity and alterations in cortisol rhythms, which can negatively impact glucose metabolism and fat storage [10,16]. These hormonal disruptions may contribute to the increased risk of obesity and metabolic syndrome observed in adolescents who habitually skip breakfast. Therefore, the physiological impact of breakfast omission extends beyond energy intake patterns and plays a role in long-term metabolic regulation.

The relationship between overweight/obesity and breakfast skipping should also consider the caloric expenditure associated with physical activity (PA), a factor often overlooked in studies addressing breakfast omission [2,3,16]. The interplay between breakfast habits and PA is complex and has been examined in multiple investigations. While some studies report no direct association between breakfast omission and sedentary behavior or PA levels [17], others suggest that PA—particularly moderate to vigorous physical activity (MVPA)—may attenuate the association between skipping breakfast and excess weight. For example, López-Gil et al. [2] observed that although adolescents who skip breakfast are at higher risk of obesity, those who engage in higher levels of daily MVPA are less likely to be overweight, regardless of their breakfast consumption. This moderating effect may be due to the role of PA in increasing energy expenditure, regulating appetite, suppressing hunger-stimulating hormones, and enhancing insulin sensitivity [5].

Therefore, it seems that the association between skipping breakfast and obesity does not necessarily imply a direct causal relationship. Other factors—such as a sedentary lifestyle, genetic predisposition, or general dietary habits—may also contribute to the development of obesity in individuals who skip breakfast. Lifestyle factors may also play a role in the decision to skip breakfast. Studies have shown that adolescents who sleep less, spend more time in front of screens, or engage in low levels of PA are more likely to skip breakfast [2,18].

Finally, the adolescent’s family, school, and social environment may also be linked to breakfast omission. Family environment and socioeconomic status can influence a teenager’s decision to skip breakfast. Parental overweight is associated with higher scores for “diet” and “snack” patterns in children, and maternal educational level is linked to children’s dietary patterns [19]. This suggests that parental eating habits can influence those of their children, including breakfast omission. Schools also play an important role in shaping adolescents’ dietary habits. The availability of school breakfast programs, nutrition education, and the creation of an environment that encourages breakfast consumption may positively influence adolescents’ decisions to eat breakfast [20].

Despite the increasing attention to breakfast habits among adolescents, several gaps remain in the literature. Many studies fail to simultaneously consider sociodemographic, psychosocial, dietary, and behavioral factors within a gender-stratified framework. Additionally, limited evidence exists on how these variables interact with adherence to the Mediterranean diet in representative adolescent populations. This study aims to address these gaps by providing a comprehensive analysis of breakfast omission and its associated factors—disaggregated by gender—using a stratified and representative sample of Spanish adolescents. By doing so, it contributes to a better understanding of this multifactorial behavior and supports the development of more tailored public health interventions.

## 2. Materials and Methods

### 2.1. Sample

The sample consisted of third-year students in compulsory secondary education (Educación Secundaria Obligatoria, ESO) from urban areas (localities with more than 10,000 inhabitants) and non-urban areas (localities with 10,000 or fewer inhabitants) across the five provinces that make up the region of Castilla-La Mancha (Toledo, Ciudad Real, Albacete, Cuenca, and Guadalajara). To ensure the representativeness of the sample, a stratified sampling method was used based on population size, taking into account students attending schools in each setting. This stratification was adopted to capture potential differences in lifestyle and dietary behaviors, as previous studies have shown that adolescents living in rural or non-urban environments may have distinct patterns of breakfast consumption compared to their urban peers [21].

The decision to focus on third-year ESO students (14–15 years old) was based on the fact that this is a critical age group for promoting healthy habits, as adolescents at this stage have greater autonomy in decision-making.

### 2.2. Inclusion and Exclusion Criteria

Inclusion criteria: Adolescents enrolled in the third year of ESO at schools selected to participate in the study, with written informed consent provided by their parents or legal guardians.

Exclusion criteria: Adolescents with physical and/or psychological conditions (as confirmed by their parents) that could limit their ability to participate in data collection.

### 2.3. Instruments

1–Questionnaire

Once informed consent was obtained, adolescents completed a questionnaire on sociodemographic, psychosocial, and lifestyle variables. The questions in this survey are part of the PACO and PACA projects, whose protocols have been previously described [22].

Lifestyle-Related Variables:Adherence to the Mediterranean diet: Assessed using the KIDMED questionnaire [23], which comprises 16 dichotomous items (YES/NO). Four items indicate lower adherence to the Mediterranean diet (consumption of fast food, pastries, sweets, and skipping breakfast), and twelve items indicate higher adherence (consumption of olive oil, fish, fruits, vegetables, cereals, nuts, legumes, pasta or rice, dairy products, and yogurt). Scores range from −1 to +1 depending on the negative or positive connotation of each item. The total score was dichotomized to simplify interpretation: good adherence (scores 7–12) and poor adherence (scores 0–6). This categorization allowed for clearer identification of adherence levels.Breakfast frequency was evaluated using the following question from the PACO Project [24]: “From Monday to Friday during the school year, how many days do you have breakfast?” Participants could choose “5 days”, “4 days”, “3 days”, “2 days”, “1 day”, or “I never eat breakfast on school days”. Students who selected “5 days” were considered regular breakfast consumers. Although breakfast consumption is part of the overall Mediterranean diet adherence index, in this study, both the total KIDMED score and individual items were analyzed as explanatory variables in relation to the main outcome: skipping breakfast (defined independently based on reported weekday frequency).Sleep duration: Questions related to sleep habits were based on the Sleep Habits Survey for Adolescents [25]. Participants indicated the exact time they went to sleep and woke up on both school days and weekends. Sleep duration was calculated based on this information.Screen time: Measured using the Screen-Time Sedentary Behavior Questionnaire (64), employed in the HELENA study [26] and the PASOS study [27]. Participants reported screen time on weekdays. From these data, two dichotomous variables were derived: total screen time (less than or more than 4 h) and adherence to screen time recommendations (less than or more than 2 h per day).Physical activity (PA): Assessed through stages of change based on the Prochaska and DiClemente model [28], which includes five stages: precontemplation, contemplation, preparation, action, and maintenance. The first three were classified as non-compliance with physical activity. Enjoyment of PA was also assessed using a Likert-type scale (none, little, somewhat, quite a bit, very much) [24]. The mode of commuting to school was also assessed as active or motorized.Well-being: Quality of life was assessed using the EQ-5D-Y-3L questionnaire from the EuroQol Group in its youth version, which includes five dimensions related to health (mobility, self-care, usual activities, pain/discomfort, and worry/sadness), with three response levels (“No problems”, “Some problems”, “A lot of problems”) [29]. For the purpose of breakfast analysis, only the “pain/discomfort” and “worry/sadness” dimensions were considered.Family and environmental variables: Monthly household income was categorized as less than EUR 1000; EUR 1000–EUR 1999; EUR 2000–EUR 2999; and EUR 3000 or more. Residential environment was classified as urban (living in a municipality with 10,000 or more inhabitants) or non-urban (fewer than 10,000 inhabitants). The usual dinner and bedtime was also noted.


2–Anthropometric Measurements: BMI


Weight was measured using a TANITA MC-780 bioimpedance scale. Height was measured using a SECA stadiometer.

Body mass index (BMI) classification into normal weight and overweight/obesity was performed according to the criteria of Cole et al. [30].

### 2.4. Statistical Analysis

Statistical analysis was conducted using SPSS version 29.0 (SPSS Inc., Chicago, IL, USA). Descriptive statistics were applied, including means, standard deviations, medians, minimum and maximum values, percentages, and absolute values, depending on whether the variables were quantitative or qualitative.

For quantitative variables, normality was assessed using the main laws of random variable distribution. Given the sample size, the Kolmogorov–Smirnov test was applied.

A bivariate analysis was performed to examine associations between the variables and the habit of skipping breakfast using Pearson’s chi-squared test and calculating the odds ratio (OR) with its corresponding 95% confidence interval (CI). A *p*-value less than 0.05 was considered statistically significant.

Finally, a multivariate analysis was conducted using binary logistic regression, with all relevant factors and the variable “breakfast: yes or no”. The aim was to develop a predictive model for breakfast omission using SPSS’s backward stepwise selection method. The model’s predictive capacity was assessed by calculating the area under the ROC curve (AUC). According to Swets’ criteria, AUC values were interpreted as follows: 0.5–0.6 (poor), 0.6–0.7 (fair), 0.7–0.8 (satisfactory), 0.8–0.9 (good), and 0.9–1.0 (excellent) [31].

Due to gender differences in breakfast consumption patterns among adolescents, statistical analyses were stratified by gender. In our sample, significant differences were observed in both the prevalence of breakfast skipping and the reasons cited for it. Therefore, performing separate analyses for boys and girls allowed us to identify gender-specific associations and predictors, ensuring a more accurate and context-sensitive interpretation of the data.

### 2.5. Ethical Considerations

This study was approved by the Ethics Committee of Castilla-La Mancha (ID: C-392). It was conducted in accordance with the principles outlined in the Declaration of Helsinki (1964), revised in Fortaleza (2013). After random selection of the participating schools, initial contact was made via email. An official presentation and informational dossier about the project were sent to each school. Follow-up phone calls were then made to arrange explanatory meetings with school leadership teams.

## 3. Results

### 3.1. Descriptive Characteristics

Table 1 presents the descriptive characteristics of the 547 adolescents studied. The gender distribution was similar (280 girls and 267 boys), with a mean age of 14.81 years. A total of 33.46% of students reported skipping breakfast on one or more weekdays. When disaggregated by gender, the rate was significantly higher among girls (43.3%) than among boys (24.4%). A bivariate analysis of the variables in Table 1 revealed several statistically significant differences between adolescents who skipped breakfast on one or more weekdays and those who did not. Girls were significantly more likely than boys to skip breakfast (*p* < 0.001). Adolescents who skipped breakfast were also more likely to report screen time exceeding 4 h per day (*p* = 0.009) and to sleep less than 8 h per night (*p* = 0.030). Finally, low adherence to the Mediterranean diet was significantly associated with skipping breakfast (*p* < 0.001).

### 3.2. Breakfast Habits by Gender

Table 2 analyzes specific characteristics related to breakfast habits, including the reasons for skipping breakfast and the number of days on which breakfast is skipped. The data analysis revealed significant gender differences in breakfast behavior: girls were more likely than boys to skip breakfast every day (14% vs. 7%, *p* < 0.001). Additionally, there were significant differences between genders in the self-reported reasons for skipping breakfast. Girls more frequently cited a lack of time (39% vs. 26%, *p* < 0.001), forgetting to eat breakfast (23% vs. 14%, *p* = 0.006), and a lack of appetite (49% vs. 29%, *p* < 0.001), whereas these reasons were not significant among boys. The data also indicate gender differences in the consumption of certain food groups for breakfast; for instance, girls were less likely than boys to have a dairy product (71% vs. 82%, *p* = 0.002) or cereals/grains (51% vs. 59%, *p* = 0.048) for breakfast.

### 3.3. Bivariate and Multivariate Analysis by Gender

To gain a deeper understanding of the factors associated with breakfast omission, bivariate and multivariate analyses were conducted separately for girls (Table 3) and boys (Table 4).

These analyses explored the relationship between the main variable of interest—skipping breakfast—and a range of sociodemographic, psychosocial, and lifestyle factors, including overall adherence to the Mediterranean diet (measured via the total KIDMED score) and specific items such as fruit, vegetable, and olive oil consumption.

#### 3.3.1. Female Adolescents

Among the 275 girls surveyed, 119 (43.3%) reported skipping breakfast. Bivariate analysis (Table 3) revealed a significant association between breakfast omission and low adherence to the Mediterranean diet (*p* < 0.001). Significant relationships were also found with hours of sleep (*p* = 0.049), the EuroQol “worry/sadness” dimension (*p* = 0.016), the EuroQol “pain/discomfort” dimension (*p* = 0.016), and bedtime (*p* = 0.033). Specifically, 62% of girls who skipped breakfast had low adherence to the Mediterranean diet compared to 42% among those who ate breakfast. Emotionally, girls who reported not feeling worried or sad were less likely to skip breakfast (60% vs. 45%).

**Table 3 nutrients-17-01948-t003:** Bivariate and multivariate analysis in female adolescents.

			Bivariate Analysis		Multivariate Analysis	
	Do Not Skip Breakfast *n* (%)	Skip Breakfast *n* (%)	OR 95% CI	*p*-Value X^2^	OR 95% CI	*p*-Value
Environment						
Urban	68 (44)	50 (42)	1.000	0.445		
Non-urban	88 (56)	69 (58)	1.066 (066–1.73)			
Monthly household income						
Less than EUR 1000	21 (13)	12 (10)	1.000	0.484		
EUR 1000-EUR 1999	67 (43)	59 (50)	1.541 (0.70–3.40)			
EUR 2000-EUR 2999	28 (18)	20 (17)	1.250 (0.50–3.11)			
More than EUR 3000	23 (15)	12 (10)	0.913 (0.34–2.47)			
EQ5D worry/sadness						
No problems	93 (60)	54 (45)	1.000	0.016 *	1	0.026 *
Some problems	59 (38)	55 (46)	1.605 (0.98–2.64)		1.651 (0.85–3.21)	
A lot of problems	4 (3)	10 (8)	4.306 (1.29–14.40)		6.438 (1.53–27.04)	
EQ5D pain/discomfort						
No problems	129 (83)	82 (69)	1.000	0.016 *		
Some problems	26 (17)	33 (28)	1.997 (1.11–3.58)			
A lot of problems	1 (1)	4 (3)	6.293 (0.69–57.29)			
Enjoy physical activity (PA)						
None	2 (1)	6 (5)	1.000	0.351		
Little	4 (3)	5 (4)	0.417 (0.05–3.31)			
Somewhat	47 (30)	30 (25)	0.213 (0.04–1.12)			
Quite a bit	59 (38)	44 (37)	0.249 (0.05–1.29)			
Very much	44 (28)	34 (29)	0.258 (0.05–1.36)			
PA recommendations						
Does not meet	52 (33)	43 (36)	1.000	0.360		
Meets	104 (67)	76 (64)	0.628 (0.54–1.46)			
Commuting to school						
Active	89 (57)	75 (63)	1.000	0.168		
Motorized	67 (43)	43 (36)	0.762 (0.47–1.24)			
Screen time						
Less than 4 h	93 (60)	59 (50)	1.000	0.062		
4 h or more	63 (40)	60 (50)	1.501 (0.93–2.43)			
Screen time recommendations						
Less than 2 h/day	30 (19)	19 (16)	1.000	0.295		
2 h/day or more	126 (81)	100 (84)	0.798 (0.42–1.50)			
Dinner time						
Before 21 pm	54 (35)	82 (69)	1.000	0.056		
Between 21:00 and 21:30	57 (37)	33 (28)	0.474 (0.26–0.88)			
Between 21:30 and 22:00	32 (21)	4 (3)	1.055 (0.56–1.98)			
After 22:00 pm	13 (8)	0 (0)	1.038 (0.43–2.49)			
Bedtime						
Before 22:00 pm	20 (13)	48 (40)	1.000	0.033 *		
Between 22:00 and 23:00	72 (46)	24 (20)	0.526 (0.25–1.11)			
Between 23:00–24:00	34 (22)	30 (25)	0.774 (0.34–1.75)			
After 24:00 pm	30 (19)	12 (10)	1.298 (0.58–2.86)			
Sleeping hours						
Less than 8 h	49 (31)	49 (41)	1.000	0.049 *		
8 h or more	107 (69)	68 (57)	0.636 (0.39–1.05)			
BMI (Cole)						
Normal weight	117 (75)	82 (69)	1.000	0.122		
Overweight/Obesity	37 (24)	37 (31)	1.427 (0.84–2.44)			
KidMed						
High adherence	65 (42)	74 (62)	1.000	<0.001 *	1	<0.001 *
Low adherence	91 (58)	45 (38)	0.434 (0.27–0.71)		0.140 (0.06–0.34)	
Kidmed (1.000 = yes)						
I have a dairy product for breakfast	136 (87)	59 (50)	0.145 (0.08–0.26)	0.001 *	0.238 (0.12–0.48)	<0.001 *
I have cereals or grains for breakfast	88 (56)	53 (45)	0.621 (0.38–1.00)	0.034 *		
I have commercially baked goods or pastries for breakfast	60 (39)	41 (34)	0.841 (0.51–1.38)	0.289		
I consume a fruit or a fruit juice every day	93 (60)	83 (70)	1.562 (0.94–2.59)	0.054	2.893 (1.34–6.23)	0.007 *
I have a second fruit everyday	64 (41)	55 (46)	1.235 (0.76–2.0)	0.230		
I have fresh or cooked vegetables regularly once a day	102 (65)	81 (68)	1.128 (0.68–1.87)	0.368	3.618 (1.59–8.24)	0.002 *
I have fresh or cooked vegetables regularly more than once a day	57 (37)	61 (51)	1.827 (1.12–2.97)	0.010 *	2.677 (1.29–5.55)	0.008 *
I consume fish regularly (2–3 times/week)	104 (67)	74 (62)	0.822 (0.50–1.35)	0.260		
I go more than once a week to a fast-food (hamburger) restaurant	31 (20)	28 (24)	1.241 (0.70–2.21)	0.279		
I like pulses and eat them more than once a week	119 (76)	87 (73)	0.845 (0.49–1.46)	0.322		
I consume pasta or rice almost every day (5 or more times/week)	72 (46)	70 (59)	1.667 (1.03–2.70)	0.025 *		
I consume nuts regularly (2–3 times per week)	78 (50)	56 (47)	0.889 (0.55–1.43)	0.359		
We use olive oil at home	153 (98)	106 (89)	0.160 (0.40–0.58)	0.002 *	0.145 (0.03–0.67)	0.014 *
I eat two yogurts and/or some cheese daily	129 (83)	82 (69)	0.464 (0.26–0.82)	0.006 *		
I eat sweets and candy several times every day	42 (27)	41 (34)	1.427 (0.85–2.40)	0.112		

* *p* < 0.005.

In the multivariate analysis, low adherence to the Mediterranean diet (OR = 0.140; 95% CI: 0.06–0.34; *p* < 0.001) and low olive oil consumption (OR = 0.145; 95% CI: 0.03–0.67; *p* = 0.014) were significant predictors of skipping breakfast. Furthermore, consuming fresh vegetables more than once per day was associated with a higher likelihood of skipping breakfast (OR = 2.677, *p* = 0.008).

#### 3.3.2. Male Adolescents

Among the 262 boys surveyed, 64 (24.4%) reported skipping breakfast. In this group, bivariate analysis (Table 4) also revealed a significant association between breakfast omission and low adherence to the Mediterranean diet (*p* < 0.001). Boys with low adherence had a significantly higher probability of skipping breakfast (77% vs. 39%). Additionally, screen time greater than 4 h per day was significantly associated with skipping breakfast (*p* = 0.015).

**Table 4 nutrients-17-01948-t004:** Bivariate and multivariate analysis in male adolescents.

			Bivariate Analysis		Multivariate Analysis	
	Do Not Skip Breakfast *n* (%)	Skip Breakfast *n* (%)	OR (95% CI)	*p*-Value X^2^	OR 95% CI	*p*-Value
Environment						
Urban	95 (48)	29 (45)	1.000	0.411		
Non-urban	103 (52)	35 (55)	1.113 (0.63–1.96)			
Monthly household income						
Less than EUR 1000	15 (8)	10 (16)	1.000	0.379		
EUR 1000–EUR 1999	72 (36)	22 (34)	0.458 (0.18–1.16)			
EUR 2000–EUR 2999	53 (27)	18 (28)	0.509 (0.20–1.33)			
More than EUR 3000	27 (14)	8 (13)	0.444(0.14–1.37)			
EQ5D worry/sadness						
No problems	157 (79)	53 (83)	1.000	0.631		
Some problems	37 (19)	9 (14)	0.721 (0.33–1.59)			
A lot of problems	4 (2)	2 (3)	1.481 (0.26–8.32)			
EQ5D pain/discomfort		0 (0)				
No problems	174 (88)	50 (78)	1.000	0.052		
Some problems	24 (12)	13 (20)	1.885 (0.90–3.97)			
A lot of problems	0 (0)	1 (2)	-			
Enjoy physical activity (PA)						
None	0 (0)	0 (0)		0.070		
Little	8 (4)	1 (2)	1.000			
Somewhat	24 (12)	16 (25)	5.333 (0.61–46.85)			
Quite a bit	70 (35)	22 (34)	2.514 (0.298–21.23)			
Very much	96 (48)	25 (39)	2.083 (0.25–17.44)			
PA recommendations						
Does not meet	32 (16)	14 (22)	1.000	0.195		
Meets	166 (84)	50 (78)	0.688 (0.34–1.39)			
Modo desplazamiento						
Activo	121 (61)	36 (56)	1.000	0.278		
Motorizado	76 (38)	28 (44)	1.238 (0.70–2.19)			
Screen time						
Less than 4 h	107 (54)	24 (38)	1.000	0.015 *		
4 h or more	91 (46)	40 (63)	1.960 (1.10–3.49)			
Screen time recommendations						
Less than 2 h/day	35 (18)	5 (8)	1.000	0.038 *		
2 h/day or more	163 (82)	59 (92)	0.395 (0.15–1.06)			
Dinner time						
Before 21 pm	85 (43)	23 (36)	1.000	0.723		
Between 21:00 and 21:30	57 (29)	20 (31)	1.297 (0.65–2.58)			
Between 21:30 and 22:00	40 (20)	13 (20)	1.201 (0.55–2.61)			
After 22:00 pm	15 (8)	7 (11)	1.725 (0.63–4.73)			
Bedtime						
Before 22:00 pm	38 (19)	9 (14)	1.000	0.014 *		
Between 22:00 and 23:00	83 (42)	18 (28)	0.916 (0.38–2.22)			
Between 23:00 and 24:00	42 (21)	12 (19)	1.206 (0.46–3.18)			
After 24:00 pm	35 (18)	23 (36)	2.775 (1.13–6.80)			
Sleeping hours						
Less than 8 h	59 (30)	21 (33)	1.000	0.324		
8 h or more	139 (70)	41 (64)	0.829 (0.45–1.52)			
BMI (Cole)						
Normal weight	136 (70)	36 (56)	1.000	0.047 *	1	0.035 *
Overweight/obesity	59 (30)	27 (42)	1.729 (0.96–3.10)		2.185 (1.06–4.52)	
KidMed						
High adherence	78 (39)	49 (77)	1.000	<0.001 *	1	<0.001 *
Low adherence	120 (61)	15 (23)	0.199 (0.10–0.38)		0.136 (0.06–0.32)	
Kidmed (1.000 = yes)						
I have a dairy product for breakfast	176 (89)	38 (59)	0.183 (0.09–0.36)	0.001 *	0.263 (0.12–0.60)	0.02 *
I have cereals or grains for breakfast	127 (64)	27 (42)	0.408 (0.23–0.73)	0.002 *		
I have commercially baked goods or pastries for breakfast	87 (44)	23 (36)	0.716 (0.40–1.28)	0.163	0.413 (0.20–0.88)	0.022 *
I consume a fruit or a fruit juice every day	116 (59)	32 (50)	0.707 (0.40–1.25)	0.145		
I have a second fruit everyday	99 (50)	22 (34)	0.524 (0.29–0.94)	0.020 *		
I have fresh or cooked vegetables regularly once a day	126 (64)	33 (52)	0.608 (0.34–1.08)	0.059		
I have fresh or cooked vegetables regularly more than once a day	79 (40)	23 (36)	0.845 (0.47–1.52)	0.340		
I consume fish regularly (2–3 times/week)	121 (61)	31 (48)	0.598 (0.34–1.05)	0.051		
I go more than once a week to a fast-food (hamburger) restaurant	46 (23)	11 (17)	0.686 (0.33–1.42)	0.201		
I like pulses and eat them more than once a week	153 (77)	41 (64)	0.524 (0.29–0.96)	0.029 *		
I consume pasta or rice almost every day (5 or more times/week)	115 (58)	33 (52)	0.768 (0.44–1.35)	0.221	2.009 (0.94–4.32)	0.074
I consume nuts regularly (2–3 times per week)	106 (54)	34 (53)	0.984 (0.56–1.73)	0.534		
We use olive oil at home	185 (93)	53 (83)	0.339 (0.14–0.80)	0.014 *		
I eat two yogurts and/or some cheese daily	152 (77)	43 (67)	0.620 (0.33–1.45)	0.088		
I eat sweets and candy several times every day	48 (24)	13 (20)	0.797 (0.40–1.59)	0.321	0.462 (0.19–1.12)	0.086

* *p* < 0.005.

In the multivariate analysis, low adherence to the Mediterranean diet (OR = 0.136; 95% CI: 0.06–0.32; *p* < 0.001) and not consuming industrial pastries (OR = 0.413; 95% CI: 0.20–0.88; *p* = 0.022) were significant predictors of skipping breakfast. Moreover, being overweight or obese (BMI) was also significantly associated with breakfast omission (OR = 2.185, *p* = 0.035).

### 3.4. Predictive Capacity of the Models

The ROC (receiver operating characteristic) curves for the female model (Figure 1, AUC = 0.807) and the male model (Figure 2, AUC = 0.792) demonstrate satisfactory predictive capacity for identifying adolescents who skip breakfast.

## 4. Discussion

This study aimed to identify the factors associated with breakfast skipping in a representative stratified sample of adolescents from the rural and urban areas from the region of Castilla-La Mancha in Spain. The analysis revealed that 33.46% of the students did not eat breakfast daily. Moreover, significant gender differences were observed, with girls being more likely than boys to skip this meal. In both genders, there was an association between breakfast omission and low adherence to the Mediterranean diet, shorter sleep duration, greater screen time, and being classified as overweight or obese.

Our findings are consistent with prior research indicating a higher prevalence of breakfast skipping among girls than among boys, as observed in studies conducted in both European and non-European adolescent populations [2,8,14]. This trend has been linked to greater body image concerns and dieting behaviors among girls, which may influence meal patterns and breakfast avoidance. Moreover, the strong association we identified between breakfast skipping and low adherence to the Mediterranean diet aligns with results from other Spanish and Mediterranean cohort studies, which suggest that adolescents who skip breakfast tend to follow a lower-quality diet overall [5,10]. Similar to our findings, Ramsay et al. [9] reported that breakfast omission is associated with a reduced intake of nutrient-dense foods such as dairy products and cereals. In contrast, the unexpected higher intake of fruits and vegetables among girls who skipped breakfast in our sample has been less commonly reported, although Pedersen et al. [32] found that meal regularity was independently associated with fruit and vegetable intake in some cases. This discrepancy may reflect differences in perceived health behaviors or diet misreporting, particularly among adolescent girls. Regarding the relationship between breakfast skipping and overweight/obesity in boys, our results support meta-analyses that have established breakfast as a protective factor against adiposity [14,33], though this association appears to be moderated by other lifestyle factors such as physical activity, which in our sample did not show a significant relationship. Finally, the association observed between breakfast skipping and indicators of emotional distress in girls is consistent with previous evidence linking this habit to higher risks of depressive symptoms, lower psychological well-being, and greater emotional instability [15,34]. These converging findings underline the multifaceted nature of breakfast skipping and reinforce the need for comprehensive, gender-sensitive public health strategies.

The prevalence of breakfast skipping in our study was approximately 34%, with a clear gender difference: 43.3% in girls versus 24.4% in boys. Previous research has reported highly variable rates (1.3% to 74.7%) [3], largely due to differing definitions of what constitutes “skipping breakfast”. Some studies assess frequency over specific timeframes (e.g., survey day, prior week, or year) [11], while others use categorical groupings (e.g., 0–2 days, 3–4 days per week) [2] or define it based on caloric intake or specific food consumption [7]. In our study, adolescents were classified as breakfast skippers if they did not eat breakfast on at least one school day; by contrast, the stricter definition of skipping every day yielded a prevalence of 11% (14% in girls, 7% in boys). These methodological differences hinder cross-study comparisons and highlight the need for consistent criteria. Among girls, the most cited reasons for skipping breakfast were lack of time, forgetfulness, and absence of appetite. Gender-stratified multivariate models in our study revealed distinct predictors of breakfast omission, reinforcing the idea that contributing factors differ by sex and supporting the need for gender-specific interventions. This aligns with prior findings that girls are more likely to skip breakfast and that such behavior may relate to other risk factors such as diet quality and overweight [2]. In both genders, breakfast skipping was strongly associated with low adherence to the Mediterranean diet.

Although breakfast skipping is one of the items included in the KIDMED questionnaire [23], we treated it as the primary outcome of interest and analyzed its association with both the total KIDMED score and individual dietary components. This approach was justified because the aim of the study was not to evaluate the questionnaire itself but to use it as a tool to explore various aspects of adolescent dietary patterns. While the overall score reflects general adherence to the Mediterranean diet, individual items provide insight into specific behaviors, such as fruit, vegetable, or olive oil consumption. Analyzing these components separately helped to avoid collinearity and to identify dietary patterns associated with breakfast omission beyond its contribution to the total score. From a public health standpoint, understanding whether breakfast skipping is linked to poor overall diet quality or to specific dietary deficits is essential for identifying risk profiles and informing targeted nutrition education strategies.

Specifically, in our study, low adherence to the Mediterranean diet and low olive oil consumption were significant predictors of breakfast skipping in the multivariate analysis for girls and also appeared in the bivariate analysis for boys. These results are supported by evidence linking higher adherence to the Mediterranean diet with regular breakfast consumption in adolescents [10]. This connection may reflect overall lower diet quality among those who skip breakfast. In fact, results from both genders showed that those who frequently skipped breakfast tended not to consume dairy products or cereals when they did eat breakfast. Surprisingly, girls who consumed more fruits and vegetables were more likely to skip breakfast, while for boys, consuming a second fruit was negatively associated with breakfast omission. A similar pattern was observed for vegetable consumption in girls. Furthermore, boys who consumed industrial pastries were less likely to skip breakfast—possibly because pastries were a common component of their breakfast. Although counterintuitive at first glance, this result may reflect specific breakfast patterns in this subgroup. It is possible that industrial pastries represent a common and accessible breakfast item for many adolescents, particularly boys, and their absence may indicate breakfast omission rather than substitution with healthier options. Alternatively, this association could be influenced by dietary reporting bias, where participants who skip breakfast or are more health-conscious may underreport the consumption of less nutritious foods such as pastries. These findings suggest that the relationship between breakfast skipping and the consumption of fruits, vegetables, or pastries may vary by gender and warrants cautious interpretation. While breakfast skipping is not specifically linked to low intake of fruits and vegetables, it is associated with poor overall diet quality [5]. However, some studies indicate that these associations may be moderated by gender or age [32]. In our study, girls may have misunderstood what constitutes a healthy diet. They might believe that increasing fruit and vegetable intake is sufficient, while being unaware of—or misinterpreting—other parameters of good Mediterranean diet adherence. They might also perceive breakfast skipping as a form of intermittent fasting believed to be health-promoting. Although intermittent fasting may be beneficial for obesity management in adolescents, evidence is limited [35]; moreover, it has been associated with disordered eating behaviors [36]. Our study did not assess whether students practiced any form of intermittent fasting. Therefore, more research is needed to explore adolescents’ perceptions of what is healthy and evaluate whether those perceptions are accurate. Adolescents may believe they are improving their health when they are not. Nutrition education in schools is essential to reinforce healthy eating patterns. Additional studies are needed to better understand adolescents’ views on healthy eating.

Another notable result in adolescent girls was the association between breakfast skipping and emotional well-being. Among girls who skipped breakfast, 54% reported feeling sad or worried. Several studies have examined the link between breakfast omission and mental health, particularly feelings of sadness or depression. Evidence suggests a significant association between breakfast skipping and a higher risk of depressive symptoms. A systematic review and meta-analysis by Zahedi et al. [15] found a positive association between skipping breakfast and an increased likelihood of depression and psychological distress across various age groups. Similarly, a study by Pengpid et al. [34], which surveyed university students from 28 countries, found that skipping breakfast was associated with several negative mental health indicators, including depression, lower happiness, and loneliness. These findings indicate that breakfast skipping is linked to a broader range of mental health issues beyond sadness and suggest that promoting breakfast consumption may improve mental well-being.

In boys, the predictive model also included BMI, showing an association between skipping breakfast and being overweight or obese. This relationship has been documented in previous studies involving children and adolescents [33], highlighting breakfast as a protective factor against cardiometabolic risk. Breakfast has also been associated with other protective factors such as lower body fat and improved insulin sensitivity [37,38]. Promoting regular breakfast consumption may be a key strategy to prevent obesity and improve overall adolescent health. Higher physical activity levels could also mitigate some of the negative effects of breakfast omission on body weight. While the relationship between breakfast skipping and PA remains unclear, PA as a healthy habit may independently confer benefits. Some studies suggest PA acts independently of breakfast habits [17], while others propose PA as an important factor to consider [16]. In our study, neither meeting PA recommendations, enjoyment of PA, nor the mode of commuting to school (active vs. motorized) was associated with breakfast habits. More extensive monitoring studies may be needed. It would also be useful to assess not only the amount and type of activity but also adolescents’ physical fitness and whether it is affected by meal-skipping behaviors.

Moreover, breakfast omission tends to co-occur with other unhealthy behaviors, such as increased screen time and reduced sleep duration. Excessive screen use—especially at night—may disrupt circadian rhythms and reduce sleep due to nighttime light exposure, which in turn has been linked to a higher likelihood of skipping breakfast [39,40]. In our study, students who skipped breakfast reported higher screen time and fewer hours of sleep, although these associations were not always significant across genders, nor were they included in the predictive models. Objective measurements of sleep and screen time may be necessary to better assess these relationships.

This study also examined family environment and socioeconomic status as potential influences on adolescents’ decisions to skip breakfast. No significant associations were found with either type of setting or household income. Although adolescents from lower socioeconomic backgrounds are often more likely to skip breakfast [41], further research is needed to explore this dimension. This pattern may be related to an obesogenic environment characterized by limited access to healthy food, lower levels of nutrition education at home, and less structured family routines, all of which can promote unhealthy eating behaviors from an early age [42].

This study has several methodological strengths worth highlighting. First, stratified sampling was employed, ensuring the representativeness of the adolescent sample from different settings (urban, semi-urban, and rural). This methodological approach offers a more accurate and generalizable view of breakfast habits among the adolescent population studied. In addition, multivariate analysis was conducted, allowing for the identification of factors independently associated with breakfast omission while controlling for potential confounding variables.

However, it is important to acknowledge certain limitations. The main limitation of this study is its cross-sectional design, which prevents the establishment of causal relationships between the variables studied and breakfast omission. Although significant associations were identified, it is not possible to determine whether skipping breakfast is the cause or the consequence of other factors, such as low adherence to the Mediterranean diet or high screen time. Additionally, data on breakfast habits and other lifestyle factors were obtained through self-reported questionnaires, which may be subject to recall bias, potentially affecting the accuracy of the results. Finally, another important limitation of this study is the relatively small sample size, which may restrict the generalizability of the findings to broader adolescent populations. Future research with larger and more diverse samples is recommended to validate and expand upon these results.

The findings of this study have important practical implications for promoting healthy breakfast habits in adolescents. Given the high prevalence of breakfast skipping (34.1%), it is crucial to implement targeted interventions that encourage breakfast consumption in this age group.

An effective strategy could involve implementing or improving school breakfast programs. The availability of school breakfast options, nutrition education, and the creation of an environment that encourages breakfast consumption may positively influence adolescents’ decisions to eat breakfast.

Another key recommendation is the promotion of nutrition education both at school and at home. Adolescents should receive clear and accurate information about the im-portance of breakfast for health. This education should address common myths and misconceptions about breakfast and offer practical ideas for preparing quick, healthy, and appealing meals.

Given the significant gender differences in breakfast habits, it is essential to design gender-specific interventions. For example, programs targeting girls could focus on overcoming barriers related to lack of time and morning appetite, while those targeting boys could emphasize reducing screen time. In both groups, promoting greater adherence to the Mediterranean diet should be a priority.

This cross-sectional study provides valuable insights into the factors associated with breakfast omission in adolescents. However, longitudinal studies are needed to better understand the causal relationship between skipping breakfast and health outcomes. Intervention studies are also necessary to evaluate the effectiveness of various strategies for promoting breakfast consumption in adolescents. Such studies could compare different types of school breakfast programs, nutrition education interventions, and personalized approaches to encourage healthy breakfast habits.

## 5. Conclusions

This study reveals that skipping breakfast is a common practice among Spanish adolescents, affecting one-third of the sample, and it is significantly more prevalent among girls than boys. A strong association was found between breakfast omission and low adherence to the Mediterranean diet, longer screen time, shorter sleep duration, and being classified as overweight or obese in both genders. In girls, a lower intake of olive oil and emotional distress were significant predictors, while in boys, being overweight/obese and the absence of industrial pastries in the diet were relevant factors.

## Figures and Tables

**Figure 1 nutrients-17-01948-f001:**
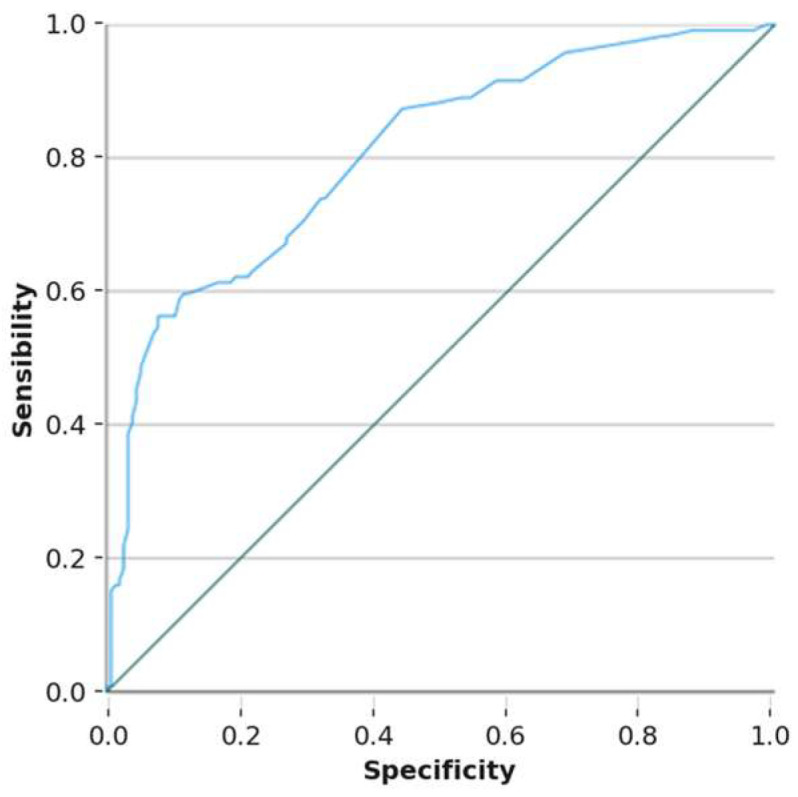
ROC curve for the female model. Area under the ROC curve to determine the predictive ability of the model in girls, representing sensitivity on the ordinate axis and 1-specificity on the abscissa.

**Figure 2 nutrients-17-01948-f002:**
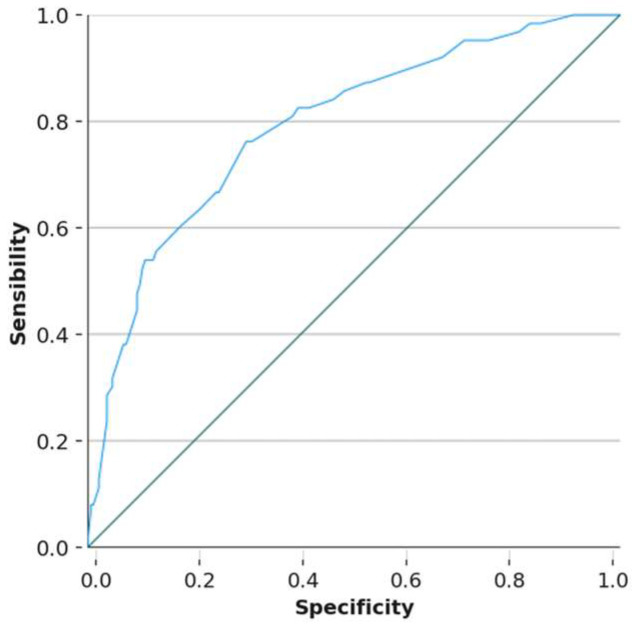
ROC curve for the male model. Area under the ROC curve to determine the predictive ability of the model in boys, representing the sensitivity on the ordinate axis and 1-specificity on the abscissa.

**Table 1 nutrients-17-01948-t001:** Descriptive characteristics.

	Total		Do Not Skip Breakfast (183)	Skip Breakfast (354)	*p*-Value
	*n* (%)	Media ± SD	*n* (%)	*n* (%)	
Gender					
Female	280 (51)		156 (44)	119 (65)	<0.001 *
Male	267 (49)		198 (56)	64 (35)	
Age		14.81 ± 0.60			0.088
Enjoy PA					
None	8 (1)		2 (1)	6 (3)	0.058
Little	18 (3)		12 (3)	6 (3)	
Somewhat	121 (22)		71 (20)	46 (25)	
Quite a bit	197 (36)		129 (36)	66 (36)	
Very much	200 (37)		140 (40)	59 (32)	
Screen time					
Less than 4 h	287 (53)		200 (56)	83 (45)	0.014 *
4 h or more	257 (47)		154 (44)	100 (55)	
Screen time recommendations					
Less than 2 h/day	453 (83)		289 (82)	159 (87)	0.121
2 h/day or more	91 (17)		65 (18)	24 (13)	
Sleeping hours		8.16 ± 0.96			
Less than 8 h	178 (33)		108 (31)	70 (39)	0.047 *
8 h or more	355 (67)		246 (69)	109 (61)	
Monthly household income					
Less than EUR 1000	14 (3)		36 (12)	22 (14)	0.573
EUR 1000-EUR 1999	13 (2)		139 (45)	81 (50)	
EUR 2000-EUR 2999	38 (7)		81 (26)	38 (24)	
More than EUR 3000	133 (25)		50 (16)	20 (12)	
BMI (Cole)		21.83 ± 4.26			
Normal weight	378 (70)		253 (72)	118 (65)	0.068
Overweight/obesity	163 (30)		96 (28)	64 (35)	
KidMed		6.23 ± 2.72			
High adherence	271 (50)		211 (60)	60 (33)	<0.001 *
Low adherence	270 (50)		143 (40)	123 (67)	
Environment					
Urban	264 (44)		163 (46)	79 (43)	0.573
Non-urban	334 (56)		191 (54)	104 (57)	

* *p* < 0.005.

**Table 2 nutrients-17-01948-t002:** Breakfast information.

	Total *n* (%)	Male *n* (%)	Female *n* (%)	Chi^2^
Breakfast 0 days	57 (11)	18 (7)	39 (14)	<0.001 *
Breakfast 1 day	19 (4)	6 (2)	13 (5)	
Breakfast 2 days	34 (6)	11 (4)	23 (8)	
Breakfast 3 days	42 (8)	13 (5)	29 (11)	
Breakfast 4 days	31 (6)	16 (6)	15 (5)	
Breakfast 5 days	354 (66)	198 (76)	156 (57)	
I do not have time for breakfast	175 (33)	67 (26)	108 (39)	<0.001 *
I do not like breakfast	81 (15)	35 (13)	46 (17)	0.166
I forget to have breakfast	100 (19)	37 (14)	63 (23)	0.006 *
I am not hungry	211 (39)	75 (29)	136 (49)	<0.001 *
I have a dairy product for breakfast	409 (76)	214 (82)	195 (71)	0.002 *
I have cereals or grains for breakfast	295 (55)	154 (59)	141 (51)	0.048 *
I have commercially baked goods or pastries for breakfast	211 (39)	110 (42)	101 (37)	0.123

* *p* < 0.005.

## Data Availability

The data and materials used for the current study are available from the corresponding author upon reasonable request.

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
