# Peer review of "Why Do Adolescents Skip Breakfast? A Study on the Mediterranean Diet and Risk Factors"

_nutrients, 2025, doi:10.3390/nu17121948_

Round 1

Reviewer 1 Report

Comments and Suggestions for Authors

Dear Authors

This study examined to a study on the Mediterranean diet and risk factors. It has been well designed and written in this study. And it is interesting and excellent study which presents good data on this important topic. I believe this is good issue in field of nutrition section.

Minor concerns

Abstract

Line 21: Please add prevalence of breakfast skipping in world statistic.

Line 30-33: Please revise all results to two decimal places in percentage.

Line 30-38: Please add the OR, 95% CI, and p value in each result, respectively.

Introduction: well-written

Line 55: contributing to weight gain[8].; there should be a space before and after these mathematical symbols: ±, =, <, >, ≤, ≥, +, −, ÷, ×, ·, ≈, ∼, ∩, ∫, Π, Σ, and |.

Methods: well-written

It is good that (1) this study investigated explanatory variables in relation to the main outcome, (2) it has inclusion and exclusion criteria.

Results

In Tables and main text, for example, Table 4, located line ‘Urban’, change from ‘1’ to ‘1.000’; as you know, two-digit numbers in mean, standard deviation, etc., three-digit numbers in statistical values (t, F value) are generally spelled out in academic writing. Please change in whole manuscript.

Discussion

Please add some more limitations such as ‘this study has small sample numbers’ in this study.

Checking by the iThenticate system, the plagiarism rate is 25% (quotes included and bibliography excluded). It is also serious flaws. You have to reduce the plagiarism rate under 10~15%.

Author Response

We thank the reviewers for their time and effort to improve our manuscript. 
We will answer all the comments individually below.
R: reviewer
A: authors
Reviewer 1
Minor concerns
Abstract
R: Line 21: Please add prevalence of breakfast skipping in world statistic.
A: Thank you for your comment. An update on the prevalence of skipped breakfast among adolescents worldwide has been added to the summary.
R: Line 30-33: Please revise all results to two decimal places in percentage.
A: Thank you for your feedback. All percentages presented in the abstract results have been revised and modified to two decimal places, ensuring uniformity and accuracy in the presentation of the data.
R: Line 30-38: Please add the OR, 95% CI, and p value in each result, respectively.
A: Thank you for your comments. Odds Ratio (OR) values, 95% confidence intervals (95% CI) and corresponding p-values have been added for each key outcome presented in the summary. 
Introduction: well-written
R: Line 55: contributing to weight gain[8].; there should be a space before and after these mathematical symbols: ±, =, <, >, ≤, ≥, +, −, ÷, ×, ·, ≈, ∼, ∩, ∫, Π, Σ, and |.
A: Thank you for your review. This has been corrected.
Methods: well-written
R: It is good that (1) this study investigated explanatory variables in relation to the main outcome, (2) it has inclusion and exclusion criteria.
A: We appreciate the positive assessment of the methodological design of the study.
Results
R: In Tables and main text, for example, Table 4, located line ‘Urban’, change from ‘1’ to ‘1.000’; as you know, two-digit numbers in mean, standard deviation, etc., three-digit numbers in statistical values (t, F value) are generally spelled out in academic writing. Please change in whole manuscript.
 A: All statistical values in the tables and in the main text have been updated to follow suggested academic conventions.
We have changed the value ‘1’ to ‘1,000’ in all corresponding lines of tables 3 and 4.
We have adjusted decimal places in means and standard deviations to two decimal places; and statistical values to three decimal places.
Discussion
R: Please add some more limitations such as ‘this study has small sample numbers’ in this study.
A: Thank you very much for your input. The following sentence has been added in the Limitations section acknowledging the small sample size as a limitation of the study, which may affect the generalizability of the results.
“Finally, another important limitation of this study is the relatively small sample size, which may restrict the generalizability of the findings to broader adolescent populations. Future research with larger and more diverse samples is recommended to validate and expand upon these results.
R: Checking by the iThenticate system, the plagiarism rate is 25% (quotes included and bibliography excluded). It is also serious flaws. You have to reduce the plagiarism rate under 10~15%.
A: Thank you very much for your plagiarism check. The bibliographic references have been removed from the document, and it has been tested on the COMPILATIO plagiarism detection platform. The result was 9%. A certificate has been added.

Reviewer 2 Report

Comments and Suggestions for Authors

This is an interesting cross-sectional study with adequate novelty. Some points should be addressed.

  • In the Abstract, the authors should reduce a bit the background section by increasing simultaneously the Conclusion section.
  • In the 3rd paragraph of the Introduction section, the authors should report relevant statistics for their country.
  • The 4rd paragraph of the Introduction section needs a bit more analysis.
  • In lines 78 and 79, the authors reported that "Skipping breakfast may disrupt metabolism and appetite-regulating hormones." This issue is very interesting and authors should add a bit more analysis for this issue.
  • The 6th and 7th paragraph should be merged in one paragraph. 
  • At the end of the Introduction section, the authors should describe the literature gap that their study aims to cover.
  • The sentence in lines 126-128 needs a relevant reference.
  • Concerning Table 1, a statistical analysis could be performed using chi-square test and student-t test by providing the corresponding p-value. A desription of the derived results could be performed for the above.
  • The authors should add a justification why they statisticaly analyse breakfast habits by gender.
  • In lines 328-347, the authors should provide a full text / paragraph without numbering. This text has a high repetition rate and it could be condensed.
  • Tyhe paragraphs in lines 348-376 should also be condensed.
  • In the Discussion section, the authors should try, a bit more, to perform a comparison analysis of their results with previous evidence.
  • At the end of the Discussion section, a separate paragraph with the Strenths and the Limitations of the study should be added.
  • The Conclusions section is too long and it should be reduced in size. Some parts of the conclusion section such as the strengths and the limitations of the study should be removed to the end of the discussion section.
  • The use of bullets should be omitted in the conclusion section. Subheadings such as Practical Implications and Recommendations as well as FutureResearch should be avoided.

Author Response

We thank the reviewers for their time and effort to improve our manuscript. 
We will answer all the comments individually below.
R: reviewer
A: authors

Reviewer 2
R: This is an interesting cross-sectional study with adequate novelty. Some points should be addressed.
In the Abstract, the authors should reduce a bit the background section by increasing simultaneously the Conclusion section.
A: Thank you for your insightful comment. We have revised the Abstract to reduce the background information and have simultaneously expanded the Conclusion section to better highlight the main findings and implications of the study.
R: In the 3rd paragraph of the Introduction section, the authors should report relevant statistics for their country.
A: We appreciate this suggestion. In response, we have updated the third paragraph of the Introduction to include recent and relevant statistics on breakfast skipping among adolescents in Spain. This addition helps contextualize our study within the national setting and highlights the importance of addressing this issue locally.
The 4rd paragraph of the Introduction section needs a bit more analysis.
Thank you for your observation. We have expanded the fourth paragraph of the Introduction to provide deeper analysis of gender-related differences in breakfast skipping. We now elaborate on the potential psychosocial and cultural factors contributing to this behavior, as well as its nutritional implications, in order to offer a more comprehensive foundation for our gender-stratified approach.
In lines 78 and 79, the authors reported that "Skipping breakfast may disrupt metabolism and appetite-regulating hormones." This issue is very interesting and authors should add a bit more analysis for this issue.
Thank you for highlighting this important point. We agree that the relationship between breakfast skipping and hormonal regulation deserves more in-depth discussion. In response, we have expanded the paragraph to incorporate additional analysis of the physiological mechanisms involved, supported by relevant literature.
R: The 6th and 7th paragraph should be merged in one paragraph. 
A: Thank you for your suggestion. We have merged the 6th and 7th paragraphs of the Introduction into a single, cohesive paragraph. The revised paragraph better integrates the discussion on physical activity (PA), its interaction with breakfast omission, and its potential role in moderating associated health risks.
R: At the end of the Introduction section, the authors should describe the literature gap that their study aims to cover.
A: We appreciate the reviewer’s insightful comment. In response, we have added a final paragraph to the Introduction that explicitly outlines the literature gap addressed by our study. This addition emphasizes the need for gender-specific analysis of breakfast skipping in adolescents, particularly in relation to Mediterranean diet adherence and lifestyle factors, using a representative sample in the Spanish context.
R: The sentence in lines 126-128 needs a relevant reference.
A: We thank the reviewer for pointing out the need to justify the inclusion of both urban and non-urban settings. The urban-rural divide is known to influence adolescents’ lifestyle behaviors, including dietary habits and breakfast consumption, due to differences in access to health resources, food environments, and socioeconomic factors. Including participants from both urban and non-urban areas allows for a more representative analysis of the adolescent population and provides insight into environmental influences on breakfast habits. A reference has been added to support this rationale.
R: Concerning Table 1, a statistical analysis could be performed using chi-square test and student-t test by providing the corresponding p-value. A description of the derived results could be performed for the above.
A: We appreciate the reviewer’s suggestion. As requested, we performed bivariate statistical analyses on the variables presented in Table 1 using the Chi-square test for categorical variables and the Student’s t-test for continuous variables. The corresponding p-values have been added to the table. In addition, a descriptive summary of the statistically significant findings has been included in the abstract, results, discussion and conclusion sections. These analyses revealed significant differences by breakfast status in gender, screen time, sleep duration, and adherence to the Mediterranean diet.
R: The authors should add a justification why they statisticaly analyse breakfast habits by gender.
A: We thank the reviewer for this observation. The decision to perform gender-stratified analyses was because previous research has consistently reported significant gender differences in breakfast habits among adolescents, with girls more likely than boys to skip breakfast. These differences may be driven by distinct psychosocial, behavioral, and cultural factors influencing eating behaviors in boys and girls, such as body image concerns, diet quality awareness, or appetite regulation.
In our own dataset, these gender disparities were clearly reflected: 43.3% of girls reported skipping breakfast compared to 24.4% of boys (p < 0.001). Additionally, the self-reported reasons for skipping breakfast and the specific dietary patterns also varied significantly by gender. Given these differences, we considered it methodologically appropriate to analyze boys and girls separately in the bivariate and multivariate models to avoid confounding and to better identify gender-specific predictors of breakfast skipping.
We have added this justification to the “Statistical Analysis” subsection of the Methods.
R: In lines 328-347, the authors should provide a full text / paragraph without numbering. This text has a high repetition rate and it could be condensed.
A: We appreciate the reviewer’s observation. In response, we have revised and condensed the paragraph in lines 328–347 to eliminate redundancy and improve clarity. The content has been rewritten as a continuous paragraph and has been integrated into the Discusion section to enhance the flow of the manuscript. 
R: Tyhe paragraphs in lines 348-376 should also be condensed.
A: We thank the reviewer for this suggestion. The text in lines 348–376 has already been revised and condensed in the current version of the manuscript. 
R: In the Discussion section, the authors should try, a bit more, to perform a comparison analysis of their results with previous evidence.
A: We appreciate the reviewer’s valuable comment. In response, we have expanded the Discussion section to include a more detailed comparative analysis of our findings in relation to previous studies. Specifically, we now further contextualize the gender differences observed in breakfast skipping, the association with Mediterranean diet adherence, and the role of psychosocial and behavioral factors.
R: At the end of the Discussion section, a separate paragraph with the Strengths and the Limitations of the study should be added.
A: We thank the reviewer for this helpful suggestion. In accordance with the recommendation, we have added a separate paragraph at the end of the Discussion section that outlines the main strengths and limitations of the study.
R: The Conclusions section is too long and it should be reduced in size. Some parts of the conclusion section such as the strengths and the limitations of the study should be removed to the end of the discussion section.
The use of bullets should be omitted in the conclusion section. Subheadings such as Practical Implications and Recommendations as well as FutureResearch should be avoided.
A: We appreciate the reviewer’s suggestion. In the revised version of the manuscript, we have removed the bullet points and restructured the conclusion into a coherent, continuous paragraph format. Additionally, we have eliminated the subheadings “Strengths and Limitations”, “Practical Implications and Recommendations” and “Future Research”.

Reviewer 3 Report

Comments and Suggestions for Authors

This study presents a study examining breakfast habits among Spanish adolescents. The topic is highly relevant given increasing concerns about adolescent nutrition and health behaviors.
Including both urban and rural adolescents adds representativeness to the findings.
Ambiguity in "breakfast skipping one or more days": This phrasing lacks clarity. Does it mean skipping breakfast at least once a week, or more frequently? The criteria for defining "skipping" should be clearly stated. Inconsistency in breakfast skipping rates: The prevalence is first described as 33.46%, then more detailed rates (43.3% for girls, 24.4% for boys) are given. These should be better integrated to avoid seeming contradictory. 
The p-value is reported as “p < 0.001” for the comparison of daily breakfast skipping between genders. While this suggests a strong difference, including exact p-values (when available) and effect sizes (e.g., odds ratios) would strengthen interpretation.
The abstract appropriately describes the study as cross-sectional, but phrases like “predictors” may imply causality. Using “associated factors” or “correlates” would be more appropriate.
"Not eating industrial pastries" as a protective factor for boys is counterintuitive and should be clarified. Was this a proxy for different eating patterns or possibly misinterpreted due to dietary reporting bias?
While differences between boys and girls are noted, the conclusion could offer more insight into why these differences exist and how interventions could address them, rather than simply stating their need.

Author Response

We thank the reviewers for their time and effort to improve our manuscript. 
We will answer all the comments individually below.
R: reviewer
A: authors

R: This study presents a study examining breakfast habits among Spanish adolescents. The topic is highly relevant given increasing concerns about adolescent nutrition and health behaviors.
Including both urban and rural adolescents adds representativeness to the findings.
Ambiguity in "breakfast skipping one or more days": This phrasing lacks clarity. Does it mean skipping breakfast at least once a week, or more frequently? The criteria for defining "skipping" should be clearly stated. Inconsistency in breakfast skipping rates: The prevalence is first described as 33.46%, then more detailed rates (43.3% for girls, 24.4% for boys) are given. These should be better integrated to avoid seeming contradictory. 
A: We thank the reviewer for the insightful feedback and positive remarks regarding the relevance and representativeness of our study. To address the concerns raised:
Clarification of the breakfast skipping definition: We have clarified the operational definition in both the Methods and Discussion sections. Specifically, "skipping breakfast" was defined as not eating breakfast on one or more school days during the week, based on self-reported weekday frequency. This corresponds to a categorization aimed at capturing any degree of irregularity in breakfast consumption.
Integration of prevalence figures: To improve clarity and consistency, we have revised the presentation of breakfast skipping rates in the Results and Discussion. The global prevalence of 33.46% refers to adolescents who skipped breakfast on at least one school day. We now explicitly state that this figure includes 43.3% of girls and 24.4% of boys, to ensure that these percentages are interpreted as subsets of the total.
R: The p-value is reported as “p < 0.001” for the comparison of daily breakfast skipping between genders. While this suggests a strong difference, including exact p-values (when available) and effect sizes (e.g., odds ratios) would strengthen interpretation.
The abstract appropriately describes the study as cross-sectional, but phrases like “predictors” may imply causality. Using “associated factors” or “correlates” would be more appropriate.
A: We agree that the term “predictors” can suggest causality if used broadly in observational research. However, in our case, the term refers specifically to the output of multivariate logistic regression models, which were validated through ROC curve analysis. Therefore, we use “predictors” strictly in a statistical sense—to refer to variables included in models with predictive purposes—and avoid any implication of causal inference. Where applicable, we have ensured that terms such as “associated factors” or “correlates” are used when discussing bivariate relationships or descriptive findings outside the predictive framework.

R: "Not eating industrial pastries" as a protective factor for boys is counterintuitive and should be clarified. Was this a proxy for different eating patterns or possibly misinterpreted due to dietary reporting bias?
A: We appreciate the reviewer’s observation regarding this counterintuitive finding. In the multivariate analysis, not consuming industrial pastries appeared as a variable associated with a higher likelihood of skipping breakfast among boys. One possible explanation is that, in this subgroup, pastries may constitute a frequent and culturally accepted component of a typical breakfast. Thus, boys who do not consume pastries might be those who forgot breakfast altogether, rather than replacing pastries with healthier alternatives. Another potential factor is dietary reporting bias: participants may underreport the consumption of less healthy items such as pastries due to social desirability, particularly among those with irregular breakfast habits. Finally, this association might also reflect broader differences in breakfast composition and routines not fully captured by the questionnaire. We have now included this clarification in the Discussion section to acknowledge the complexity of this finding and the need for further research to explore the role of breakfast composition and reporting accuracy.
R: While differences between boys and girls are noted, the conclusion could offer more insight into why these differences exist and how interventions could address them, rather than simply stating their need.
A: We thank the reviewer for this insightful recommendation. In response, we have revised the conclusion to incorporate a more interpretive perspective on the observed gender differences. We now discuss potential underlying factors, such as differing motivations, habits, and perceptions related to diet and health, and suggest how interventions could be adapted to better address these specific needs. 

Round 2

Reviewer 2 Report

Comments and Suggestions for Authors

The authors have significantly revised and improved their manuscript.

Reviewer 3 Report

Comments and Suggestions for Authors

After the major modifications made in the article, I think it is publishable.